# GANITE: Estimation of Individualized Treatment Effects using Generative Adversarial Nets

**Jinsung Yoon**
Department of Electrical and Computer Engineering
University of California, Los Angeles
Los Angeles, CA 90095, USA
`jsyoon0823@g.ucla.edu`

**James Jordon**
Department of Engineering Science
University of Oxford
Oxford, UK
`james.jordon@wolfson.ox.ac.uk`

**Mihaela van der Schaar**
Department of Engineering Science, University of Oxford, Oxford, UK
Alan Turing Institute, London, UK
`mihaela.vanderschaar@eng.ox.ac.uk`

## Abstract

Estimating individualized treatment effects (ITE) is a challenging task due to the need for an individual's potential outcomes to be learned from biased data and without having access to the counterfactuals. We propose a novel method for inferring ITE based on the Generative Adversarial Nets (GANs) framework. Our method, termed Generative Adversarial Nets for inference of Individualized Treatment Effects (GANITE), is motivated by the possibility that we can capture the uncertainty in the counterfactual distributions by attempting to learn them using a GAN. We generate proxies of the counterfactual outcomes using a counterfactual generator, $\mathbf{G}$, and then pass these proxies to an ITE generator, $\mathbf{I}$, in order to train it. By modeling both of these using the GAN framework, we are able to infer based on the factual data, while still accounting for the unseen counterfactuals. We test our method on three real-world datasets (with both binary and multiple treatments) and show that GANITE outperforms state-of-the-art methods.

## 1 Introduction

Individualized treatment effects (ITE) estimation using observational data is a fundamental problem that is applicable in a wide variety of domains. For instance, (1) in understanding the heterogeneous effects of drugs (Shalit et al. (2017); Alaa & van der Schaar (2017); Alaa et al. (2017)); (2) in evaluating the effect of a policy on unemployment rates (LaLonde (1986); Smith & Todd (2005)); (3) in verifying which factor causes a certain disease (Höfler (2005)) and (4) estimating the effects of pollution on the weather (Hannart et al. (2016)).

As explained in Spirtes (2009), the problem of ITE estimation differs from the standard supervised learning problem. First, among the potential outcomes, only the factual outcome is actually observed (revealed), counterfactual outcomes are not observed and so the entire vector of potential outcomes can never be obtained. Second, unlike randomized controlled trials (RCT), observational studies are prone to treatment selection bias. For instance, left ventricular assist device (LVAD) treatment is mostly applied to high-risk patients with severe cardiovascular diseases before heart transplantation, the distribution of features among these patients will be significantly different to the distribution among non-LVAD treated patients (Kirklin et al. (2010)). The sample distribution can vary drastically across different choices of treatments and therefore, if we were to apply a supervised learning framework for each treatment separately, the learned models would not generalize well to the entire population.

Classical works in this domain solved the problem of estimating the average treatment effects from observational data (Dehejia & Wahba (2002b); Lunceford & Davidian (2004)). These works account

for the selection bias using propensity scores (the estimated probability of receiving a treatment) to create unbiased estimators of the average treatment effect. Dehejia & Wahba (2002b) used a one-to-one matching methodology to pair treated and control patients with similar features while Lunceford & Davidian (2004) used propensity scoring weighing to account for the selection bias. More recent works focus on individualized treatment effects (Chipman et al. (2010); Wager & Athey (2017); Athey & Imbens (2016); Lu et al. (2017); Alaa & van der Schaar (2017); Porter et al. (2011); Johansson et al. (2016); Alaa et al. (2017); Louizos et al. (2017); Shalit et al. (2017)). Detailed qualitative comparisons to these works will be discussed in the next subsection and numerical comparisons can be found in Section 5.

In this paper, we propose a novel approach that attempts to not only fit a model to the observed factual data, but also account for the unseen counterfactual outcomes. We view the factual outcome as an observed label and consider the counterfactual outcomes to be missing labels. Missing labels are generated by the well-known Generative Adversarial Nets (GAN) framework (Goodfellow et al. (2014)). More specifically, the counterfactual generator of GANITE attempts to generate counterfactual outcomes in such a way that when given the combined vector of factual and generated counterfactual outcomes the discriminator of GANITE cannot determine which of the components is the factual outcome. With the complete labels (combined factual and estimated counterfactual outcomes), the ITE estimation function can then be trained for inferring the potential outcomes of the individual based on the feature information in a supervised way. By also modelling this ITE estimation function using a GAN framework, we are able not only to predict the expected outcomes but also provide confidence intervals for the predictions, which is very important in, for example, the medical setting.

Unlike many other state-of-the-art methods, our method naturally extends to - and in fact is defined in the first place for - any number of treatments. We conduct experiments with three real-world observational datasets (with both binary and multiple treatments), and GANITE outperforms state-of-the-art methods.

## 1.1 RELATED WORKS

Previous works on ITE estimation can be divided into three categories. In the first, a separate model is learned for each treatment; this approach does not account for selection bias and so each model learned will be biased toward the distribution of that treatment's population. In the second, the treatment is considered a feature, with one model learned for everything, and the mismatch between the entire sample distribution and treated and control distributions is adjusted in order to account for selection bias. For instance, Chipman et al. (2010); Wager & Athey (2017); Athey & Imbens (2016); Lu et al. (2017) used tree-based models, Porter et al. (2011) used doubly-robust methods, Dehejia & Wahba (2002b); Lunceford & Davidian (2004), k-nearest neighbor (kNN) Crump et al. (2008) used propensity and matching based methods, and Johansson et al. (2016); Shalit et al. (2017) used deep learning approaches to solve the ITE problem under this one model methodology. Inherent to the approach of learning a balanced representation is that the representation must trade off between containing predictive information and reducing biased information. This is because often it will be the case that information that is biased is also highly predictive (in fact in the medical setting this is precisely why it is biased - because the doctors will assign treatments based on predictive features). On the other hand, our framework is not forced to make this information trade-off - the dataset we learn our final ITE estimator on contains the original dataset, and so contains at least as much information as that one. In the experiment section, we show that our proposed framework outperforms Shalit et al. (2017), particularly when the bias is high. In the third category, Alaa & van der Schaar (2017); Alaa et al. (2017) used a multi-task model approach. Alaa et al. (2017) used multi-task neural nets to estimate (1) the selection bias, (2) the controlled outcome and (3) the treated outcome with shared layers across these three tasks. Alaa & van der Schaar (2017) used a Gaussian Process approach in the multi-task model setting. Our work is perhaps most similar to Alaa & van der Schaar (2017) since there, too, they attempted to account for the counterfactuals and were similarly able to provide confidence in their estimates using credible intervals. They were able to access counterfactuals through a posterior distribution which was then accounted for in the learning of their model.

## 2 PROBLEM FORMULATION: ESTIMATION OF INDIVIDUALIZED TREATMENT EFFECTS

Let $\mathcal{X}$ denote the $s$-dimensional feature space and $\mathcal{Y}$ the set of possible outcomes. Consider a joint distribution, $\mu$, on $\mathcal{X} \times \{0, 1\}^k \times \mathcal{Y}^k$ where $k$ is the number of possible treatments. Suppose that $(\mathbf{X}, \mathbf{T}, \mathbf{Y}) \sim \mu$. We call $\mathbf{X} \in \mathcal{X}$ the ($s$-dimensional) feature vector, $\mathbf{T} \equiv (T_1, ..., T_k) \in \{0, 1\}^k$ the treatment vector and $\mathbf{Y} \equiv (Y_1, ..., Y_T) \in \mathcal{Y}^k$ the vector of potential outcomes (or the *Individualized Treatment Effects (ITE)*). We assume that (with probability 1), there is precisely one non-zero component of $\mathbf{T}$ and we denote by $\eta$ the index of this component. Denote by $\mu_{\mathbf{X}}$ the marginal distribution of $\mathbf{X}$ and by $\mu_{\mathbf{Y}}(\mathbf{x})$ the conditional distribution of $\mathbf{Y}$ given $\mathbf{X} = \mathbf{x}$, for $\mathbf{x} \in \mathcal{X}$ (marginalized over $\mathbf{T}$). This setting is known as the Rubin-Neyman causal model (Rubin (2005)).

We introduce two[1] assumptions about the distribution $\mu$ in the Rubin-Neyman causal model.

**Assumption 1.** *(Overlap) For all $\mathbf{x} \in \mathcal{X}$, for all $i \in \{1, ..., k\}$,*
$$0 < \mathbb{P}(T_i = 1 | \mathbf{X} = \mathbf{x}) < 1.$$

This assumption ensures that at every point in the feature space, there is a non-zero probability of being given treatment $i$ for every $i$.

**Assumption 2.** *(Unconfoundedness) Conditional on $\mathbf{X}$, the potential outcomes, $\mathbf{Y}$, are independent of $\mathbf{T}$,*
$$\mathbf{Y} \perp\!\!\!\perp \mathbf{T} | \mathbf{X}.$$

This assumption is also referred to as *no unmeasured confounding* and requires that all *joint* influences on $\mathbf{Y}$ and $\mathbf{T}$ are measured. Note that this assumption means that $\mu_{\mathbf{Y}}(\mathbf{x})$ no longer needs to be marginalized over $\mathbf{T}$, since, under this assumption, they are independent.

Assume now that we observe samples of $(\mathbf{X}, \mathbf{T}, Y_\eta)$ (whose joint distribution we denote by $\mu_f$), so that our dataset, $\mathcal{D}$, is given by $\mathcal{D} = (\mathbf{x}(n), \mathbf{t}(n), y_{\eta(n)}(n))_{n=1}^N$. Importantly, we only observe the component of the potential outcome vector that corresponds to the assigned treatment, we call this the *factual* outcome, and refer to unobserved potential outcomes as counterfactual outcomes or just *counterfactuals*. We denote by $y_f(n)$ and $\mathbf{y}_{cf}(n)$ the factual outcome and (vector of) counterfactual outcome(s), respectively. From this point forward, we omit the dependence on $n$ for ease of notation.

In this setting, we wish to be able to draw samples from $\mu_{\mathbf{Y}}(\mathbf{x})$ for any $\mathbf{x} \in \mathcal{X}$. We measure the performance of the generator, $\mathbf{I}(\mathbf{x})$, using two different metrics depending on whether $k = 2$ (i.e. binary treatments) or $k > 2$ (i.e. multiple treatments).

For $k = 2$ we use the expected Precision in Estimation of Heterogeneous Effects, $\epsilon_{PEHE}$, introduced in Hill (2011), given by:
$$\epsilon_{PEHE} = \mathbb{E}_{\mathbf{x} \sim \mu_{\mathbf{X}}} \left[ \left( \mathbb{E}_{\mathbf{y} \sim \mu_{\mathbf{Y}}(\mathbf{x})}[y_1 - y_0] - \mathbb{E}_{\hat{\mathbf{y}} \sim \mathbf{I}(\mathbf{x})}[\hat{y}_1 - \hat{y}_0] \right)^2 \right]. \tag{1}$$

For $k > 2$ we use the expected mean squared error:
$$\epsilon_{MSE} = \mathbb{E}_{\mathbf{x} \sim \mu_{\mathbf{X}}} \left[ ||\mathbb{E}_{\mathbf{y} \sim \mu_{\mathbf{Y}}(\mathbf{x})}[\mathbf{y}] - \mathbb{E}_{\hat{\mathbf{y}} \sim \mathbf{I}(\mathbf{x})}[\hat{\mathbf{y}}]||_2^2 \right] \tag{2}$$
where $|| \cdot ||_2$ is the standard $\ell_2$-norm in $\mathbb{R}^k$.

In order to achieve this goal, we separate the problem into two parts. First, we attempt to generate proxies for the unobserved counterfactual outcomes using a counterfactual generator ($\mathbf{G}$) to create a *complete* dataset. Then, using this proxy dataset, we learn the ITE generator, $\mathbf{I}$.

## 3 GANITE: GENERATIVE ADVERSARIAL NETS FOR INFERENCE OF INDIVIDUALIZED TREATMENT EFFECT ESTIMATION

### 3.1 OVERVIEW

The objective of GANITE is to generate potential outcomes for a given feature vector $\mathbf{x}$. However, due to the lack of counterfactual outcomes we are unable to learn the distribution of potential outcomes directly. To account for these counterfactuals, we first attempt to generate samples, $\tilde{\mathbf{y}}_{cf}$, using

---

[1] All works cited in the Related Works section make this assumption.

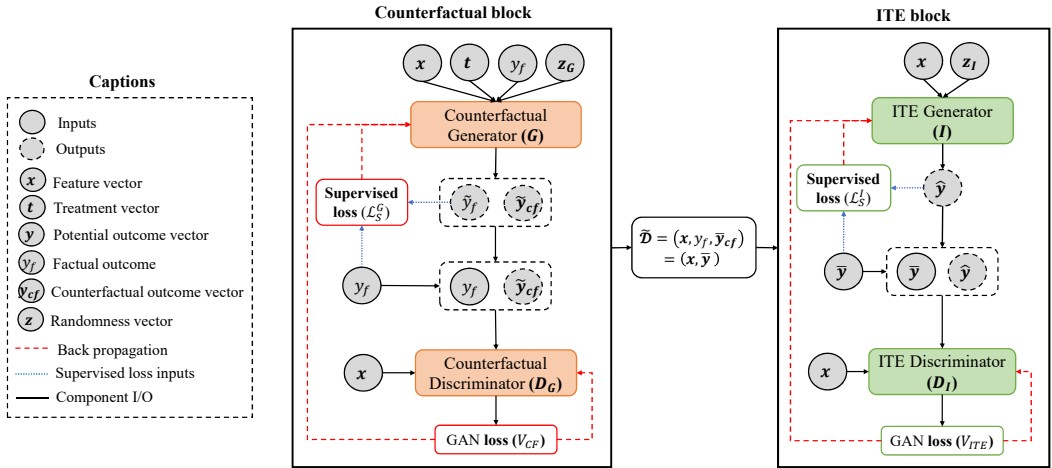

Figure 1: Block Diagram of GANITE ($\bar{\mathbf{y}}$ is sampled from $\mathbf{G}$ after $\mathbf{G}$ has been fully trained). $\mathbf{G}, \mathbf{D_G}, \mathbf{D_I}$ are only operating during training, whereas $\mathbf{I}$ operates both during training and at runtime.

a counterfactual generator, $\mathbf{G}$, from the distribution $\mu_{\mathbf{Y}_{cf}}(\mathbf{x}, \mathbf{t}, y_f)$ (the conditional distribution of the counterfactual outcomes, $\mathbf{Y}_{cf}$ given that $\mathbf{X} = \mathbf{x}$, $\mathbf{T} = \mathbf{t}$ and $Y_\eta = y_f$) for each sample in our dataset. We can then combine these *proxy counterfactuals* with the original dataset to obtain a *complete* dataset $\tilde{\mathcal{D}} = \{\mathbf{x}(n), \mathbf{t}(n), \tilde{\mathbf{y}}(n)\}_{n=1}^N$ where $\tilde{\mathbf{y}}$ is the combination of $y_f$ and $\tilde{\mathbf{y}}_{cf}$ (with $\tilde{y}_\eta = y_f$). The ITE generator, $\mathbf{I}$, can then be optimized using $\tilde{\mathcal{D}}$.

We follow a conditional GAN framework similar to the one set out in Mirza & Osindero (2014) to model the latter of these generators. For the former, we have to use a different discriminator in order to capture the same idea. More specifically, GANITE consists of two blocks: a counterfactual imputation block and an ITE block, each of which consists of a generator and a discriminator. We describe each of these blocks and their components in more detail in the following subsection.

## 3.2 A Detailed Breakdown

**Counterfactual generator (G):** The counterfactual generator, $\mathbf{G}$, uses the feature vector, $\mathbf{x}$, the treatment vector, $\mathbf{t}$, and the factual outcome, $y_f$, to generate a potential outcome vector, $\tilde{\mathbf{y}}$. We let $g$ be a function $g : \mathcal{X} \times \{0,1\}^k \times \mathcal{Y} \times [-1,1]^{k-1} \to \mathcal{Y}^k$ and $\mathbf{z_G} \sim \mathcal{U}((-1,1)^{k-1})$. We then define the random variable $\mathbf{G}(\mathbf{x}, \mathbf{t}, y_f)$ as

$$\mathbf{G}(\mathbf{x}, \mathbf{t}, y_f) = g(\mathbf{x}, \mathbf{t}, y_f, \mathbf{z_G}) \tag{3}$$

The goal now is to find a function, $g$, such that $\mathbf{G}(\mathbf{x}, \mathbf{t}, y_f) \sim \mu_{\mathbf{Y}}(\mathbf{x}, \mathbf{t}, y_f)$. We write $\tilde{\mathbf{y}}$ to denote a sample of $\mathbf{G}$ and $\bar{\mathbf{y}}$ to denote the vector obtained by replacing $\tilde{y}_\eta$ with $y_f$. Observe that the $\eta$-th component of a sample from $\mu_{\mathbf{Y}}(\mathbf{x}, \mathbf{t}, y_f)$ will be $y_f$, since we are sampling $\mathbf{Y}$ *conditional* on $Y_\eta = y_f$.

**Counterfactual discriminator ($\mathbf{D_G}$):** We introduce a discriminator, $\mathbf{D_G}$, which maps pairs $(\mathbf{x}, \bar{\mathbf{y}})$ to vectors in $[0,1]^k$ with the $i$-th component, written $D_{\mathbf{G}}(\mathbf{x}, \tilde{\mathbf{y}})_i$, representing the probability that the $i$-th component of $\tilde{\mathbf{y}}$ is the factual outcome, equivalently the probability that $\eta = i$. This is in contrast to the standard GAN framework in which the discriminator is given a single sample from one of two distributions and it attempts to determine which distribution it came from. Here the discriminator is given a sample consisting of components from two different distributions and attempts to determine which components came from which distribution.

We train $\mathbf{D_G}$ to maximize the probability of correctly identifying $\eta$. We then train $\mathbf{G}$ to maximize the probability of $\mathbf{D_G}$ *incorrectly* identifying $\eta$ (equivalently we try to minimize the probability of a correct identification - this is the adversarial method of learning between $\mathbf{G}$ and $\mathbf{D_G}$).

Following the framework in Goodfellow et al. (2014), we note that this formulation is captured by a minimax problem given by

$$\min_{\mathbf{G}} \max_{\mathbf{D_G}} \mathbb{E}_{(\mathbf{x},\mathbf{t},y_f)\sim\mu_f} \left[ \mathbb{E}_{\mathbf{z_G}\sim\mathcal{U}((-1,1)^k)} \left[ \mathbf{t}^T \log \mathbf{D_G}(\mathbf{x},\tilde{\mathbf{y}}) + (\mathbf{1}-\mathbf{t})^T \log(1 - \mathbf{D_G}(\mathbf{x},\tilde{\mathbf{y}})) \right] \right] \quad (4)$$

where $\log$ is performed element-wise and $^T$ denotes the transpose operator.

After training the counterfactual generator, we use it to generate the dataset $\tilde{\mathbf{D}}$ and pass this dataset to the ITE block.

**ITE generator (I):** The ITE generator, $\mathbf{I}$, uses only the feature vector, $\mathbf{x}$, to generate a potential outcome vector, $\hat{\mathbf{y}}$. Similar to our approach with $\mathbf{G}$, let $h$ be a function $h : \mathcal{X} \times [-1,1]^k \to \mathcal{Y}^k$ and $\mathbf{z_I} \sim \mathcal{U}((-1,1)^k)$. We define the random variable $\mathbf{I}(\mathbf{x})$ as

$$\mathbf{I}(\mathbf{x}) = h(\mathbf{x}, \mathbf{z_I}) \quad (5)$$

and similarly, the goal is to find a function, $h$, such that $\mathbf{I}(\mathbf{x}) \sim \mu_{\mathbf{Y}}(\mathbf{x})$. We write $\hat{\mathbf{y}}$ to denote a sample from $\mathbf{I}(\mathbf{x})$.

**ITE discriminator ($\mathbf{D_I}$):** Again, we introduce a discriminator, $\mathbf{D_I}$, but this time, since we have access to a *complete* dataset $\tilde{\mathbf{D}}$, we can use a standard conditional GAN discriminator - it takes a pair $(\mathbf{x}, \mathbf{y}^*)$ and returns a scalar corresponding to the probability that $\mathbf{y}^*$ was from the data $\tilde{\mathcal{D}}$ (rather than drawn from $\mathbf{I}$). Again, we train the generator and discriminator in an adversarial fashion using the following minimax criteria

$$\min_{\mathbf{I}} \max_{\mathbf{D_I}} \mathbb{E}_{\mathbf{x}\sim\mu_{\mathbf{X}}} \left[ \mathbb{E}_{\mathbf{y}^*\sim\mu_{\mathbf{Y}}}(\mathbf{x}) \left[ \log \mathbf{D_I}(\mathbf{x},\mathbf{y}^*) \right] + \mathbb{E}_{\mathbf{y}^*\sim\mathbf{I}(\mathbf{x})} \left[ \log(\mathbf{1} - \mathbf{D_I}(\mathbf{x},\mathbf{y}^*)) \right] \right] \quad (6)$$

where again $\log$ is taken element-wise.

## 4 GANITE: Optimization

In this section, we describe the empirical loss functions that are used to optimize each component of GANITE. The Pseudo-code is summarized in the Appendix.

### 4.1 Counterfactual block ($\mathbf{G}, \mathbf{D_G}$):

Based on equation 4, the empirical objective of the minimax problem for $\mathbf{G}$ and $\mathbf{D_G}$ can be defined by

$$V_{CF}(\mathbf{x}(n), \mathbf{t}(n), \bar{\mathbf{y}}(n)) = \mathbf{t}(n)^T \log(\mathbf{D_G}(\mathbf{x}(n), \bar{\mathbf{y}}(n))) + (\mathbf{1}-\mathbf{t}(n))^T \log(\mathbf{1} - \mathbf{D_G}(\mathbf{x}(n), \bar{\mathbf{y}}(n))).$$

We also introduce the following 'supervised' loss in order to enforce the restriction that $g_\eta$ should be equal to $y_f$.

$$\mathcal{L}_S^G(y_f(n), \tilde{y}_{\eta(n)}(n)) = (y_f(n) - \tilde{y}_{\eta(n)}(n))^2$$

More specifically, due to the structure of $G$, it outputs a full vector of potential outcomes, and so it not only outputs counterfactuals, but also gives a value for the one factual that was used as input. We account for this by using $\mathcal{L}_S^G$ to force the generated factual outcome to be close to the actually observed factual outcome. This is because, as noted above, conditional on observing $y_f$, the component of $y$ corresponding to $y_f$ should clearly be equal to $y_f$.

With the above two objective functions, $\mathbf{G}$ and $\mathbf{D_G}$ are iteratively optimized with $k_G$ minibatches as follows:

$$\min_{\mathbf{D_G}} -\sum_{n=1}^{k_G} V_{CF}(\mathbf{x}(n), \mathbf{t}(n), \bar{\mathbf{y}}(n))$$

$$\min_{\mathbf{G}} \sum_{n=1}^{k_G} \left[ V_{CF}(\mathbf{x}(n), \mathbf{t}(n), \bar{\mathbf{y}}(n)) + \alpha \mathcal{L}_S^G(y_f(n), \tilde{y}_{\eta(n)}(n)) \right]$$

where $\alpha \geq 0$ is a hyper-parameter.

## 4.2 ITE BLOCK ($\mathbf{I}, \mathbf{D_I}$):

After training the counterfactual block ($\mathbf{G}, \mathbf{D_G}$), GANITE optimizes the ITE block ($\mathbf{I}, \mathbf{D_I}$). Based on equation 6, the empirical objective of the minimax problem for $\mathbf{I}$ and $\mathbf{D_I}$ can be defined by

$$V_{ITE}(\mathbf{x}(n), \bar{\mathbf{y}}(n), \hat{\mathbf{y}}(n)) = \log(\mathbf{D_I}(\mathbf{x}(n), \bar{\mathbf{y}}(n))) + \log(1 - \mathbf{D_I}(\mathbf{x}(n), \hat{\mathbf{y}}(n))).$$

Furthermore, in order to optimize the performance with respect to equations 1 and 2, we additionally introduce supervised losses (for the respective cases of $k = 2$ (binary treatments) and $k > 2$ (multiple treatments)) that are defined as follows:

$$(k = 2) : \mathcal{L}_S^I(\bar{\mathbf{y}}(n), \hat{\mathbf{y}}(n)) = ((\bar{y}_1(n) - \bar{y}_0(n)) - (\hat{y}_1(n) - \hat{y}_0(n)))^2$$
$$(k > 2) : \mathcal{L}_S^I(\bar{\mathbf{y}}(n), \hat{\mathbf{y}}(n)) = ||\bar{\mathbf{y}}(n) - \hat{\mathbf{y}}(n)||_2^2.$$

$\mathbf{I}$ and $\mathbf{D_I}$ are then iteratively optimized with $k_I$ minibatches as follows:

$$\min_{\mathbf{D_I}} - \sum_{n=1}^{k_I} V_{ITE}(\mathbf{x}(n), \bar{\mathbf{y}}(n), \hat{\mathbf{y}}(n))$$

$$\min_{\mathbf{I}} \sum_{n=1}^{k_G} \left[ V_{ITE}(\mathbf{x}(n), \bar{\mathbf{y}}(n), \hat{\mathbf{y}}(n)) + \beta \mathcal{L}_S^I(\bar{\mathbf{y}}(n), \hat{\mathbf{y}}(n)) \right]$$

where $\beta \geq 0$ is a hyper-parameter.

Empirical justification for the inclusion of all the above losses can be found in Section 5.3 where we explore the effect of training with and without each of the losses, see Table 6. We demonstrate there that using a combination of both (for both $\mathbf{G}$ and $\mathbf{I}$) gives the best performance. In addition to this, by using a GAN loss for $\mathbf{I}$ we learn the conditional *distribution* of the potential outcomes rather than just the expectations (which would be the case if we only used the supervised loss $\mathcal{L}_S^I$. See Table 1 in Section 5.3). This allows us to capture the uncertainty of the outcomes, which is very important in the medical setting when treatment decisions need to be made by doctors on the basis of these types of estimations.

Due to the lack of ground truth, it is often difficult in causal inference tasks to optimize the hyper-parameters. More specifically, we do not have access to the true loss function (either PEHE or MSE) that we are trying to minimize, and so it is not possible to select hyper-parameters that minimise the *true* loss. One of the advantages of GANITE, however, is that our target loss can be estimated from the generated counterfactuals, unlike other methods such as in Shalit et al. (2017). Therefore, we can directly optimize the hyper-parameters that minimize this estimated PEHE/MSE over the hyper-parameter space - details of our hyper-parameter optimization and the achieved optimal hyper-parameters are illustrated in the Appendix.

## 5 EXPERIMENTS

### 5.1 DATASETS

Due to the nature of the problem, it is very difficult to evaluate the performance of the algorithm on real-world datasets - we never have access to the *ground truth*. Previous works, such as Shalit et al. (2017); Louizos et al. (2017), use both semi-synthetic datasets (either the treatments or the potential outcomes are synthesized) and datasets collected from randomized controlled trials (RCT) to evaluate the ITE generator. We use two semi-synthetic datasets, IHDP and Twins, and one real-world dataset, Jobs, to evaluate the performance of GANITE with various state-of-the-art methods. These datasets are the same as the ones used in Shalit et al. (2017); Louizos et al. (2017). Below, we give a detailed explanation of Twins. The details of IHDP and Jobs are well described in Shalit et al. (2017); Hill (2011); Dehejia & Wahba (2002a) and the Appendix.

**Twins:** This dataset is derived from all births in the USA between 1989-1991 (Almond et al. (2005)). Among these births, we only focus on the twins. We define the treatment $t = 1$ as being the heavier twin (and $t = 0$ as being the lighter twin). The outcome is defined as the 1-year mortality. For each twin-pair we obtained 30 features relating to the parents, the pregnancy and the birth: marital status; race; residence; number of previous births; pregnancy risk factors; quality of care

during pregnancy; and number of gestation weeks prior to birth. We only chose twins weighing less than 2kg and without missing features (list-wise deletion). This creates a complete dataset (without missing data). The final cohort is 11,400 pairs of twins whose mortality rate for the lighter twin is 17.7%, and for the heavier 16.1%. In this setting, for each twin pair we observed both the case $t = 0$ (lighter twin) and $t = 1$ (heavier twin); thus, the ground truth of individualized treatment effect is known in this dataset. In order to simulate an observational study, we selectively observe one of the two twins using the feature information (creating selection bias) as follows: $t|\mathbf{x} \sim \text{Bern}(\text{Sigmoid}(\mathbf{w}^T\mathbf{x} + n))$ where $\mathbf{w}^T \sim \mathcal{U}((-0.1, 0.1)^{30 \times 1})$ and $n \sim \mathcal{N}(0, 0.1)$.

## 5.2 PERFORMANCE METRICS AND SETTINGS

We use four different performance metrics: expected Precision in Estimation of Heterogeneous Effect (PEHE), average treatment effect (ATE) (Hill (2011)), policy risk ($R_{pol}(\pi)$), and average treatment effect on the treated (ATT) (Shalit et al. (2017)). In this subsection, we only provide definitions for PEHE and $R_{pol}(\pi)$. ATE and ATT are explained (and reported) in the Appendix.

If both factual and counterfactual outcomes are generated from a known distribution (so that we are able to compute the expectations of the outcomes, like in the IHDP dataset), and the treatment is binary, the empirical PEHE ($\epsilon_{PEHE}$) can be defined as follows:

$$\epsilon_{PEHE} = \frac{1}{N} \sum_{n=1}^{N} \Big( \mathbb{E}_{(y_1(n), y_0(n)) \sim \mu_{\mathbf{Y}}(\mathbf{x}(n))} [y_1(n) - y_0(n)] - [\hat{y}_1(n) - \hat{y}_0(n)] \Big)^2$$

where $y_1(n), y_0(n)$ are treated and controlled outcomes drawn from the ground truth ($\mu_{\mathbf{Y}}(\mathbf{x})$) and $\hat{y}_1(n), \hat{y}_0(n)$ are their estimations.

If both factual and counterfactual outcomes are observed but the underlying distribution is unknown (like in the Twins dataset), $\hat{\epsilon}_{PEHE}$ can be defined as follows:

$$\hat{\epsilon}_{PEHE} = \frac{1}{N} \sum_{n=1}^{N} \Big( [y_1(n) - y_0(n)] - [\hat{y}_1(n) - \hat{y}_0(n)] \Big)^2$$

If only factual outcomes are available but the testing set comes from a randomized controlled trial (RCT), such as in the Jobs dataset, Policy risk ($\mathcal{R}_{pol}(\pi)$) can be defined as follows (Shalit et al. (2017)):

$$R_{pol}(\pi) = \frac{1}{N} \sum_{n=1}^{N} \Big[ 1 - \Big( \sum_{i=1}^{k} \big[ \frac{1}{|\Pi_i \cap T_i \cap E|} \sum_{\mathbf{x}(n) \in \Pi_i \cap T_i \cap E} y_i(n) \times \frac{|\Pi_i \cap E|}{|E|} \big] \Big) \Big]$$

where $\Pi_i = \{\mathbf{x}(n) : i = \arg\max \hat{\mathbf{y}}\}$, $T_i = \{\mathbf{x}(n) : t_i(n) = 1\}$, and $E$ is the subset of RCT.

Each dataset is divided 56/24/20% into training/validation/testing sets. Hyper-parameters such as the number of hidden layers $\alpha$ and $\beta$ are chosen using Random Search (Bergstra & Bengio (2012)). Details about the hyper-parameters are discussed in the Appendix. We run each algorithm 100 times (except for the IHDP dataset, on which we run each algorithm 1,000 times which is the same setting in Shalit et al. (2017)) with new training/validation/testing splits and report the mean and standard deviation of the performances.

## 5.3 EXPERIMENTAL RESULTS

In our first simulation we focus on demonstrating the effect that including each of the losses introduced in Section 4 has on the performance of the algorithm. As is demonstrated below, inclusion of all four losses gives the best results.

We generate a synthetic dataset as follows: we draw 10,000 10-dimensional feature vectors $\mathbf{x} \sim \mathcal{N}(\mathbf{0}^{10 \times 1}, 0.5 \times (\Sigma + \Sigma^T))$ where $\Sigma \sim \mathcal{U}((-1, 1)^{10 \times 10})$. The treatment assignment is then generated as $t|\mathbf{x} \sim \text{Bern}(\text{Sigmoid}(\mathbf{w}_t^T\mathbf{x} + n_t))$ where $\mathbf{w}_t^T \sim \mathcal{U}((-0.1, 0.1)^{10 \times 1})$ and $n_t \sim \mathcal{N}(0, 0.1)$. The potential outcome vector is then generated as $\mathbf{y}|\mathbf{x} \sim (\mathbf{w}_y^T\mathbf{x} + \mathbf{n}_y)$ where $\mathbf{w}_y^T \sim \mathcal{U}((-1, 1)^{10 \times 2})$ and $\mathbf{n}_y \sim \mathcal{N}(\mathbf{0}^{2 \times 1}, 0.1 \times I^{2 \times 2})$. We use 8,000 instances for training and 2,000 instances for testing. We repeat this 100 times and report the average $\epsilon_{PEHE}$ on the testing set.

| | | G | | |
|---|---|---|---|---|
| | **PEHE** | **S loss only** | **GAN loss only** | **S and GAN loss** |
| **I** | **S loss only** | $.397 \pm .011$ (15.6%) | $.610 \pm .017$ (45.1%) | $.352 \pm .012$ (4.8%) |
| | **GAN loss only** | $.607 \pm .044$ (44.8%) | $.513 \pm .029$ (34.7%) | $.463 \pm .015$ (27.6%) |
| | **S and GAN loss** | $.362 \pm .011$ (7.5%) | $.491 \pm .030$ (31.8%) | $.335 \pm \mathbf{.011}$ (-) |

Table 1: Performance with various combinations of GANITE (**G**: Counterfactual generator, **I**: ITE generator, S loss: Supervised loss). Details of each cell are illustrated in the Appendix as Figures.

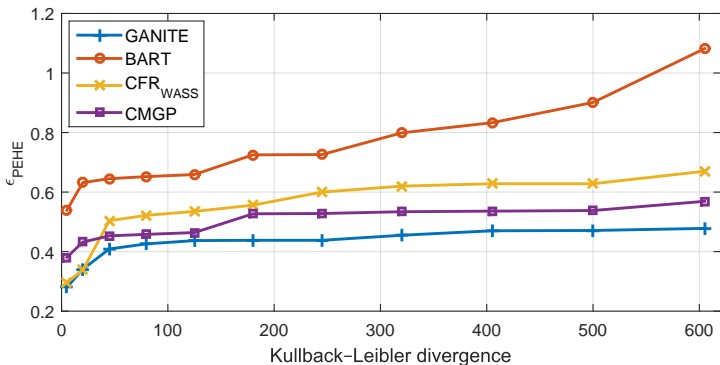

Figure 2: Performance comparison between GANITE and state-of-the-art methods as the selection bias is varied (Kullback-Leibler divergence of treated with respect to controlled distributions)

Table 1 shows the performance of the GANITE architecture using different combinations of the four losses in Section 4. For each generator component, we can use 3 different combinations of loss: (1) Supervised loss (S loss) only (this reduces the corresponding component to a standard neural network), (2) GAN loss only, (3) both S loss and GAN loss. The top-left most entry corresponds to simply using a standard neural network to first impute the counterfactuals and then using another standard neural network to learn an ITE estimator from the imputed dataset. As can be seen, this already performs well, but by adding the GAN losses for both the imputation step and the estimation step, a significant gain is shown (15.6%) (the bottom-right entry).

In Fig. 2 we show that GANITE is robust to an increased selection bias. We generate 10,000 10-dimensional treated samples from $\mathbf{x}_1 \sim \mathcal{N}(\mu_{\mathbf{1}}, 0.5 \times (\Sigma + \Sigma^T))$ and controlled samples from $\mathbf{x}_0 \sim \mathcal{N}(\mu_{\mathbf{0}}, 0.5 \times (\Sigma + \Sigma^T))$ where $\Sigma \sim \mathcal{U}((-1, 1)^{10 \times 10})$. Fixing $\mu_{\mathbf{0}}$ and varying $\mu_{\mathbf{1}}$, we generate various datasets with different Kullback-Leibler divergences (KL divergences) of $\mu_1$ with respect to $\mu_0$. A higher KL divergence indicates a higher selection bias (a larger mismatch) between treated and controlled distributions. As seen in Fig. 2, GANITE robustly outperforms state-of-the-art methods such as Shalit et al. (2017); Alaa & van der Schaar (2017) across the entire range of tested divergences.

**Binary treatments:** In this section, we evaluate GANITE for estimating individualized treatment effects for binary treatments. We use three datasets and report the $\epsilon_{PEHE}$ both in-sample and out-of-sample (for ATE and ATT see the appendix). We compare GANITE with least squares regression using treatment as a feature (OLS/LR$_1$), separate least squares regressions for each treatment (OLS/LR$_2$), balancing linear regression (BLR) (Johansson et al. (2016)), k-nearest neighbor (k-NN) (Crump et al. (2008)), Bayesian additive regression trees (BART) (Chipman et al. (2010)), random forests (RForest) (Breiman (2001)), causal forests (C Forest) (Wager & Athey (2017)), balancing neural network (BNN) (Johansson et al. (2016)), treatment-agnostic representation network (TAR-NET) (Shalit et al. (2017)), counterfactual regression with Wasserstein distance (CFR$_{WASS}$) (Shalit et al. (2017)), and multi-task gaussian process (CMGP) (Alaa & van der Schaar (2017)). We evaluate both in-sample and out-of-sample performance in Table 2.

| Methods | Datasets (Mean $\pm$ Std) | | | | | |
|---|---|---|---|---|---|---|
| | IHDP ($\sqrt{\epsilon_{PEHE}}$) | | Twins ($\sqrt{\hat{\epsilon}_{PEHE}}$) | | Jobs ($\mathcal{R}_{pol}(\pi)$) | |
| | In-sample | Out-sample | In-sample | Out-sample | In-sample | Out-sample |
| **GANITE** | $1.9 \pm .4$ | $2.4 \pm .4$ | $\mathbf{.289 \pm .005}$ | $\mathbf{.297 \pm .016}$ | $\mathbf{.13 \pm .01}$ | $\mathbf{.14 \pm .01}$ |
| OLS/LR$_1$ | $5.8 \pm .3^*$ | $5.8 \pm .3^*$ | $.319 \pm .001^*$ | $.318 \pm .007$ | $.22 \pm .00^*$ | $.23 \pm .02^*$ |
| OLS/LR$_2$ | $2.4 \pm .1$ | $2.5 \pm .1$ | $.320 \pm .002$ | $.320 \pm .003^*$ | $.21 \pm .00^*$ | $.24 \pm .01^*$ |
| BLR | $5.8 \pm .3^*$ | $5.8 \pm .3^*$ | $.312 \pm .003^*$ | $.323 \pm .018$ | $.22 \pm .01^*$ | $.25 \pm .02^*$ |
| k-NN | $2.1 \pm .1$ | $4.1 \pm .2^*$ | $.333 \pm .001^*$ | $.345 \pm .007^*$ | $.02 \pm .00$ | $.26 \pm .02^*$ |
| BART | $2.1 \pm .1$ | $2.3 \pm .1$ | $.347 \pm .009^*$ | $.338 \pm .016$ | $.23 \pm .00^*$ | $.25 \pm .02^*$ |
| R Forest | $4.2 \pm .2^*$ | $6.6 \pm .3^*$ | $.306 \pm .002^*$ | $.321 \pm .005$ | $.23 \pm .01^*$ | $.28 \pm .02^*$ |
| C Forest | $3.8 \pm .2^*$ | $3.8 \pm .2^*$ | $.366 \pm .003^*$ | $.316 \pm .011$ | $.19 \pm .00^*$ | $.20 \pm .02^*$ |
| BNN | $2.2 \pm .1$ | $2.1 \pm .1$ | $.325 \pm .003^*$ | $.321 \pm .018$ | $.20 \pm .01^*$ | $.24 \pm .02^*$ |
| TARNET | $.88 \pm .02$ | $.95 \pm .02$ | $.317 \pm .005^*$ | $.315 \pm .003$ | $.17 \pm .01^*$ | $.21 \pm .01^*$ |
| CFR$_{WASS}$ | $.71 \pm .02$ | $\mathbf{.76 \pm .02}$ | $.315 \pm .007^*$ | $.313 \pm .008$ | $.17 \pm .01^*$ | $.21 \pm .01^*$ |
| CMGP | $\mathbf{.65 \pm .44}$ | $.77 \pm .11$ | $.320 \pm .002^*$ | $.319 \pm .008$ | $.22 \pm .03^*$ | $.24 \pm .05$ |

Table 2: Performance of ITE estimation with three real-world datasets. Bold indicates the method with the best performance for each dataset. $^*$: is used to indicate methods that GANITE shows a statistically significant improvement over.

As can be seen in Table 2, GANITE achieves significant performance gains on the Twins and Jobs datasets in comparison with state-of-the-art methods (both in-sample and out-of-sample [2]). GANITE achieves a much higher gain for individualized treatment effect estimations (such as $\hat{\epsilon}_{PEHE}$ and $\mathcal{R}_{pol}(\pi)$) than average treatment effect estimations (such as $\hat{\epsilon}_{ATE}$ and $\epsilon_{ATT}$). On IHDP, GANITE is competitive with BART and BNN but is outperformed by TARNET, CFR$_{WASS}$ and CMGP. We believe this is due to the fact that GANITE has a large number of parameters to be optimized and IHDP is a relatively small dataset (747 samples). This belief is backed up by our significant gains over these methods in both Twins and Jobs, where the number of samples is much larger, (11400 and 3212 samples, respectively).

| Methods | Metric: MSE$_y$ | | | |
|---|---|---|---|---|
| | In Sample | Gain (%) | Out Sample | Gain (%) |
| **GANITE** | $\mathbf{.0427 \pm .0161}$ | (-) | $\mathbf{.0723 \pm .0183}$ | (-) |
| OLS/LR$_1$ | $.0855 \pm .0096$ | 50.1% | $.0871 \pm .0142$ | 17.0% |
| OLS/LR$_2$ | $.0857 \pm .0099$ | 50.2% | $.0883 \pm .0147$ | 18.1% |
| BLR | $.0996 \pm .0081^*$ | 57.1% | $.1017 \pm .0127$ | 28.9% |
| KNN | $.0930 \pm .0101^*$ | 54.1% | $.1008 \pm .0236$ | 28.3% |
| BART | $.1097 \pm .0084^*$ | 61.1% | $.1037 \pm .0283$ | 30.3% |
| R Forest | $.0442 \pm .0069$ | 3.4% | $.0927 \pm .0138$ | 22.0% |
| C Forest | $.1607 \pm .0014^*$ | 73.4% | $.1665 \pm .0035^*$ | 56.6% |
| BNN | $.0602 \pm .0102$ | 29.1% | $.1031 \pm .0145$ | 29.9% |
| TARNET | $.0854 \pm .0091$ | 50.0% | $.0879 \pm .0030$ | 17.7% |
| CFR$_{WASS}$ | $.0896 \pm .0036^*$ | 52.3% | $.0894 \pm .0057$ | 19.1% |
| CMGP | $.0844 \pm .0073^*$ | 49.4% | $.0793 \pm .0191$ | 8.3% |

Table 3: Performance of multiple treatment effects estimations using Twins data. Bold indicates the method with the best performance for each dataset. $^*$: is used to indicate methods that GANITE shows a statistically significant improvement over.

---

[2]Note that both "in-sample" and "out-sample" experiments evaluate the performance of **I** networks.

**Multiple treatments:** GANITE is naturally defined for estimating multiple treatment effects. In this subsection, we further preprocess the Twins data to create a dataset containing multiple treatments. The multiple treatments are determined as follows: (1) $t = 1$: lower weight, female sex, (2) $t = 2$: lower weight, male sex, (3) $t = 3$: higher weight, female sex, (4) $t = 4$: higher weight, male sex. Therefore, we have 4 possible treatments for each sample.

We use the mean-squared error:

$$\text{MSE}_y = \frac{1}{N \times |\mathcal{T}_i|} \sum_{i=1}^{N} \sum_{t \in \mathcal{T}_i} \Big( y_t(x_i) - \hat{y}_t(x_i) \Big)^2$$

as the performance metric to evaluate the multiple treatment effects (other metrics, such as PEHE, do not have a natural extension to the multiple treatments setting). For comparison with state-of-the-art methods, we naively extend BLR, C Forest, BNN, TARNET, CFR$_{WASS}$, and CMGP for multiple treatments: one of the four treatments is selected as a control treatment and then the remaining three create three separate binary ITE estimation problems (all against the same chosen control treatment).

As can be seen in Table 3 compared with Table 2 GANITE significantly outperforms other state-of-the-art methods such as TARNET and CFR$_{WASS}$ (17.7% and 19.1% gains in terms of out sample MSE, respectively). This is because, GANITE is designed for multiple treatments; the model is jointly trained for all treatments. On the other hand, other methods are designed for binary treatments and only naively extend to multiple treatments by training pairs of the available treatments.

## 5.4 DISCUSSION

The experimental results provide various intuitions of the GANITE framework for ITE estimation. First, GANITE can be easily extended to any number of treatments and performs well in this multiple treatment setting. As can be seen in Section 4, however, a different loss for binary treatment and multiple treatments must be used because PEHE is only defined for the binary treatment setting - the MSE is not a natural generalisation of the PEHE and we believe exploration of possible loss functions in this setting would be an interesting future work.

A further extension to this problem would be to consider a setting in which a patient may receive several treatments (rather than just one). While this work can handle this problem naively (by treating each *combination* of treatments as a separate 'treatment') we believe this would also be an interesting problem to explore in a future work.

## 6 CONCLUSION

In this paper we introduced a novel method for dealing with the ITE estimation problem. We have shown empirically that our method is more robust to large selection biases and performs better on standard benchmark datasets than other state-of-the-art methods. Our method also achieves significant performance gains over state-of-the-art when estimating ITE for multiple treatments because it is able to jointly estimate the representations across the multiple treatments.

## ACKNOWLEDGMENTS

This work was supported by the Office of Naval Research (ONR) and the NSF (Grant number: ECCS1462245, ECCS1533983, and ECCS1407712).

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

APPENDIX

PSEUDO-CODE OF GANITE

---
**Algorithm 1** Pseudo-code of GANITE
---

> **while** convergence of training loss of $\mathbf{G}$ and $\mathbf{D_G}$ **do**
>> **(1) Counterfactual block optimization**
>> Use $k_C$ minibatches, iteratively optimize $\mathbf{G}, \mathbf{D_G}$ by stochastic gradient descent (SGD)

$$\min_{\mathbf{D_G}} - \sum_{n=1}^{k_G} V_{CF}(\mathbf{x}(n), \mathbf{t}(n), \tilde{\mathbf{y}}(n))$$

$$\min_{\mathbf{G}} \sum_{n=1}^{k_G} \left[ V_{CF}(\mathbf{x}(n), \mathbf{t}(n), \tilde{\mathbf{y}}(n)) + \alpha \mathcal{L}_S^G(y_f(n), \tilde{y}_{\eta(n)}^*(n)) \right]$$

> **while** convergence of training loss of $\mathbf{I}$ and $\mathbf{D_I}$ **do**
>> **(2) ITE block optimization**
>> Use $k_I$ minibatches, update $\mathbf{I}, \mathbf{D_I}$ by SGD

$$\min_{\mathbf{D_I}} - \sum_{n=1}^{k_I} V_{ITE}(\mathbf{x}(n), \tilde{\mathbf{y}}(n), \hat{\mathbf{y}}(n))$$

$$\min_{\mathbf{I}} \sum_{n=1}^{k_G} \left[ V_{ITE}(\mathbf{x}(n), \tilde{\mathbf{y}}(n), \hat{\mathbf{y}}(n)) + \beta \mathcal{L}_S^I(\tilde{\mathbf{y}}(n), \hat{\mathbf{y}}(n)) \right]$$

---

DETAILED DESCRIPTION OF THE DATASETS

IHDP

Hill (2011) provided a dataset for ITE estimation with the Infant Health and Development Program (IHDP). The dataset consists of 747 children ($t = 1$: 139, $t = 0$ 608) with 25 features. We generated potential outcomes from setting A in the NPCI package Dorie (2016).

JOBS

Jobs data studied in LaLonde (1986) is composed of randomized data based on the National Supported Work program and non-randomized data from observational studies. We use a (random) subset of the randomized data to evaluate the algorithms based on $\mathcal{R}_{pol}(\pi)$ and $\epsilon_{ATT}$. The dataset consists of 722 randomized samples ($t = 1$: 297, $t = 0$: 425) and 2490 non-randomized samples ($t = 1$: 0, $t = 0$: 2490), all with 7 features.

SUMMARY OF THE DATASETS

| Data | Condition | | | | Property | | |
|---|---|---|---|---|---|---|---|
| | **F** | **CF** | **Distribution** | **RT-test** | $T$ | $N$ | $s$ |
| **IHDP** | ✓ | ✓ | Known | | Binary | 747 | 25 |
| **Jobs** | ✓ | X | Unknown | ✓ | Binary | 3212 | 7 |
| **Twins-Binary** | ✓ | ✓ | Unknown | | Binary | 11400 | 30 |
| **Twins-Multiple** | ✓ | X | Unknown | | Multiple | 11400 | 30 |

Table 4: Summary of the datasets (N is the number of samples, s is the feature-dimension)

| Methods | Datasets (Mean $\pm$ Std) | | | | | |
|---|---|---|---|---|---|---|
| | IHDP ($\epsilon_{ATE}$) | | Twins ($\hat{\epsilon}_{ATE}$) | | Jobs ($\epsilon_{ATT}$) | |
| | In-sample | Out-sample | In-sample | Out-sample | In-sample | Out-sample |
| **GANITE** | $.43 \pm .05$ | $.49 \pm .05$ | $.0058 \pm .0017$ | $.0089 \pm 0.0075$ | $\mathbf{.01 \pm .01}$ | $\mathbf{.06 \pm .03}$ |
| OLS/LR$_1$ | $.73 \pm .04^*$ | $.94 \pm .06^*$ | $\mathbf{.0038 \pm .0025}$ | $.0069 \pm .0056$ | $\mathbf{.01 \pm .00}$ | $.08 \pm .04$ |
| OLS/LR$_2$ | $.14 \pm .01$ | $.31 \pm .02$ | $.0039 \pm .0025$ | $.0070 \pm .0059$ | $\mathbf{.01 \pm .01}$ | $.08 \pm .03$ |
| BLR | $.72 \pm .04^*$ | $.93 \pm .05^*$ | $.0057 \pm .0036$ | $.0334 \pm .0092^*$ | $\mathbf{.01 \pm .01}$ | $.08 \pm .03$ |
| k-NN | $.14 \pm .01$ | $.90 \pm .05^*$ | $.0028 \pm .0021$ | $\mathbf{.0051 \pm .0039}$ | $.21 \pm .01^*$ | $.13 \pm .05$ |
| BART | $.23 \pm .01$ | $.34 \pm .02$ | $.1206 \pm .0236^*$ | $.1265 \pm .0234^*$ | $.02 \pm .00$ | $.08 \pm .03$ |
| R Forest | $.73 \pm .05^*$ | $.96 \pm .06^*$ | $.0049 \pm .0034$ | $.0080 \pm .0051$ | $.03 \pm .01$ | $.09 \pm .04$ |
| C Forest | $.18 \pm .01$ | $.40 \pm .03$ | $.0286 \pm .0035^*$ | $.0335 \pm .0083^*$ | $.03 \pm .01$ | $.07 \pm .03$ |
| BNN | $.37 \pm .03$ | $.42 \pm .03$ | $.0056 \pm .0032$ | $.0203 \pm .0071$ | $.04 \pm .01$ | $.09 \pm .04$ |
| TARNET | $.26 \pm .01$ | $.28 \pm .01$ | $.0108 \pm .0017^*$ | $.0151 \pm .0018$ | $.05 \pm .02$ | $.11 \pm .04$ |
| CFR$_{WASS}$ | $.25 \pm .01$ | $.27 \pm .01$ | $.0112 \pm .0016^*$ | $.0284 \pm .0032^*$ | $.04 \pm .01$ | $.09 \pm .03$ |
| CMGP | $\mathbf{.11 \pm .10}$ | $\mathbf{.13 \pm .12}$ | $.0124 \pm .0051$ | $.0143 \pm .0116$ | $.06 \pm .06$ | $.09 \pm .07$ |

Table 5: Performance of average treatment effect estimation. Bold represents the best performance. $^*$: statistically significant improvement of GANITE.

PERFORMANCE METRICS AND THE RESULTS OF AVERAGE TREATMENT EFFECT ESTIMATION

In this subsection, we use two different performance metrics for average treatment effect (ATE) estimation: average treatment effect (ATE) (Hill (2011)), and average treatment effect on the treated (ATT) (Shalit et al. (2017)).

If both factual and counterfactual outcomes are generated from a known distribution (and so we are able to compute the expected value, such as in IHDP), the error of ATE ($\epsilon_{ATE}$) is defined as:

$$\epsilon_{ATE} = ||\frac{1}{N}\sum_{n=1}^{N}\mathbb{E}_{\mathbf{y}(n)\sim\mu_{\mathbf{Y}}(\mathbf{x}(n))}[\mathbf{y}(n)] - \frac{1}{N}\sum_{i=1}^{n}\hat{\mathbf{y}}(n)||_2^2$$

where $\hat{\mathbf{y}}$ is the estimated potential outcome.

If both factual and counterfactual outcomes are observed but the underlying distribution is unknown (like in Twins), $\epsilon_{\hat{A}TE}$ is defined as:

$$\hat{\epsilon}_{ATE} = ||\frac{1}{N}\sum_{n=1}^{N}\mathbf{y}(n) - \frac{1}{N}\sum_{n=1}^{N}\hat{\mathbf{y}}(n)||_2^2$$

If only factual outcomes are available (such as in Jobs), treatment is binary, and the testing set comes from a randomized controlled trial (RCT), the true average treatment effect on the treated (ATT) is defined as follows (Shalit et al. (2017)):

$$ATT = \frac{1}{|T_1 \cap E|}\sum_{\mathbf{x}_i \in T_1 \cap E} Y_1(\mathbf{x}_i) - \frac{1}{|T_0 \cap E|}\sum_{\mathbf{x}_i \in C \cap E} Y_0(\mathbf{x}_i)$$

$$\epsilon_{ATT} = |ATT - \frac{1}{|T_1 \cap E|}\sum_{\mathbf{x}_i \in T_1 \cap E}\hat{Y}_1(\mathbf{x}_i) - \hat{Y}_0(\mathbf{x}_i)|$$

where $T_1$ is the subset corresponding to treated samples, $T_0$ is the subset corresponding to controlled samples, and $E$ is the subset corresponding to the randomized controlled trials.

Table. 5 shows the performance of the various algorithms with respect to these metrics.

As can be seen in Table 5, the GANITE achieves competitive performances for Average Treatment Effect (ATE) estimation but not the best model to estimate ATE (except the Jobs dataset). However, we do not believe that it is an important metric for distinguishing models where the task is predicting

treatment effects on an individual level. The problem we address with GANITE is to estimate the ITE. We used the ATE performance as a sanity check for our method - and believe it passes the sanity check, being competitive with most other methods.

HYPER-PARAMETER OPTIMIZATION

We optimize our hyper-parameters in GANITE by estimating the PEHE (MSE in the case of multiple treatments) on the dataset generated by G and minimizing this with respect to the hyper-parameters. The table below indicates specifics of this process, including the values we search over.

Table 6: Hyper-parameters of GANITE

| Blocks | Sets of Hyper-parameters |
|---|---|
| **Initialization** | Xavier Initialization for Weight matrix, Zero initialization for bias vector. |
| **Optimization** | Adam Moment Optimization |
| **Batch size** ($k_G, k_I$) | $\{32, 64, 128, 256\}$ |
| **Depth of layers** | $\{1, 3, 5, 7, 9\}$ |
| **Hidden state dimension** | $\{s, \text{int}(s/2), \text{int}(s/3), \text{int}(s/4), \text{int}(s/5)\}$ |
| $\alpha, \beta$ | $\{0, 0.1, 0.5, 1, 2, 5, 10\}$ |

For the hyper-parameter optimization of the benchmarks, we follow the hyper-parameter optimization code published in the github with their main code. For instance, the hyper-parameters of $\text{CFR}_{WASS}$ are optimized using `cfr_param_search.py` file which is published in `https://github.com/clinicalml/cfrnet`

OPTIMAL HYPER-PARAMETERS FOR EACH DATASET

Table 7: Optimal Hyper-parameters of GANITE

| Dataset | Optimal Hyper-parameters |
|---|---|
| IHDP | $k_G : 64, k_I : 64$, Depth of layers: 5, $h_{dim}$: 8, $\alpha : 2, \beta : 5$ |
| Jobs | $k_G : 128, k_I : 128$, Depth of layers: 3, $h_{dim}$: 4, $\alpha : 1, \beta : 5$ |
| Twins - Binary | $k_G : 128, k_I : 128$, Depth of layers: 5, $h_{dim}$: 8, $\alpha : 2, \beta : 2$ |
| Twins - Multiple | $k_G : 128, k_I : 128$, Depth of layers: 7, $h_{dim}$: 8, $\alpha : 1, \beta : 2$ |

ADDITIONAL EXPERIMENTS

PERFORMANCE GAP BETWEEN $G$ AND $I$

As explained in Section 4, $I$ tries to learn the potential distribution that consists of factual outcomes and counterfactual outcomes that are generated by $G$. To evaluate how well $I$ is able to learn from $G$, we compare the performance of $G$ and $I$ of the GANITE framework in Table 8 in terms of ITE estimation. As can be seen in Table 8, the in-sample ITE estimation performance of GANITE (I) is competitive with GANITE (G). It experimentally verifies that GANITE (I) learns well from the outputs of GANITE (G).

ZERO OUT THE CONTRIBUTION OF FACTUAL OUTCOME AND TREATMENT

In this section, we use the trained $G$ to compute ITE only with $x$. The learned function $G$ needs $x$, $t$, and $y_f$ as the inputs. Therefore, in order to compute ITE only with $x$, we should zero out the contribution of $t$ and $y_f$ in the $G$ function. $G$ tries to learn the conditional probability $\mathcal{P}(y|x, y_f, t)$

| Methods | Datasets (Mean $\pm$ Std) | | |
|---|---|---|---|
| | IHDP ($\sqrt{\epsilon_{PEHE}}$) | Twins ($\sqrt{\hat{\epsilon}_{PEHE}}$) | Jobs ($\mathcal{R}_{pol}(\pi)$) |
| GANITE (I) | $1.9 \pm .4$ | $.289 \pm .005$ | $.13 \pm .01$ |
| GANITE (G) | $1.4 \pm .2$ | $.267 \pm .004$ | $.10 \pm .01$ |

Table 8: Performance comparison of in-sample ITE estimation between GANITE (I) and GANITE (G) with three real-world datasets.

and what we want to compute is the conditional probability $\mathcal{P}(y|x)$. Therefore, zero out the impact of $t$ and $y_f$ can be done as follows.

$$\mathcal{P}(y|x) = \int \mathcal{P}(y|x, t, y_f)P(y_f, t|x)dtdy_f$$

The $\mathcal{P}(y|x, t, y_f)$ is learned by $G$ and $P(y_f, t|x)$ can be easily learned using supervised learning framework (all the labels $(y_f, t)$ are available) with drop-out approach (to approximate the integral as the sample mean of multiple samples). We use multi-layer perceptron (MLP) with multiple outputs to learn the function $P(y_f, t|x)$. We called this as zero-out GANITE. We compare the performance of zero-out GANITE to original GANITE in Table 9. As can be seen in Table 9, the performance of original GANITE is marginally better than zero-out GANITE in three different datasets. We do not believe that zero-out GANITE would be any simpler than our proposed structure - in both cases we would still have 2 learning stages (Table 9 experimentally verifies this).

| Methods | Datasets (Mean $\pm$ Std) | | | | | |
|---|---|---|---|---|---|---|
| | IHDP ($\sqrt{\epsilon_{PEHE}}$) | | Twins ($\sqrt{\hat{\epsilon}_{PEHE}}$) | | Jobs ($\mathcal{R}_{pol}(\pi)$) | |
| | In-sample | Out-sample | In-sample | Out-sample | In-sample | Out-sample |
| GANITE | $1.9 \pm .4$ | $2.4 \pm .4$ | $.289 \pm .005$ | $.297 \pm .016$ | $.13 \pm .01$ | $.14 \pm .01$ |
| zero-out GANITE | $2.2 \pm .6$ | $2.6 \pm .7$ | $.297 \pm .008$ | $.308 \pm .022$ | $.14 \pm .02$ | $.17 \pm .01$ |

Table 9: Performance comparison of ITE estimation between GANITE and zero-out GANITE with three real-world datasets.

TOY EXAMPLE

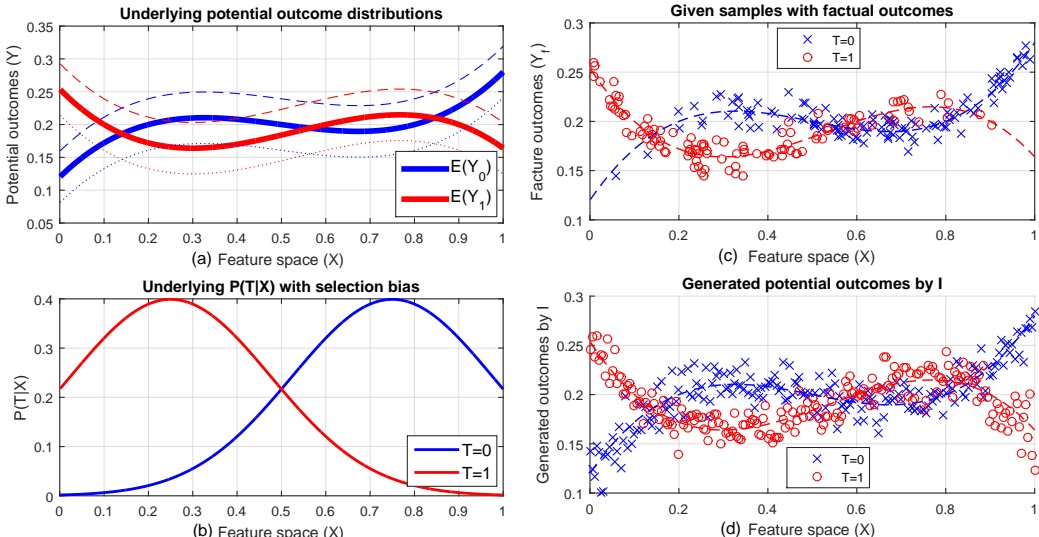

Figure 3: (a) Underlying distribution of potential outcomes (**Y**), (b) Underlying distribution of treatment assignments $\mathcal{P}(T|\mathbf{X})$, (c) Training data (factual outcomes) sampled from distributions explained in (a) and (b), (d) Potential outcomes sampled from trained ITE generator (**I**).

ADDITIONAL FIGURES FOR TABLE 1

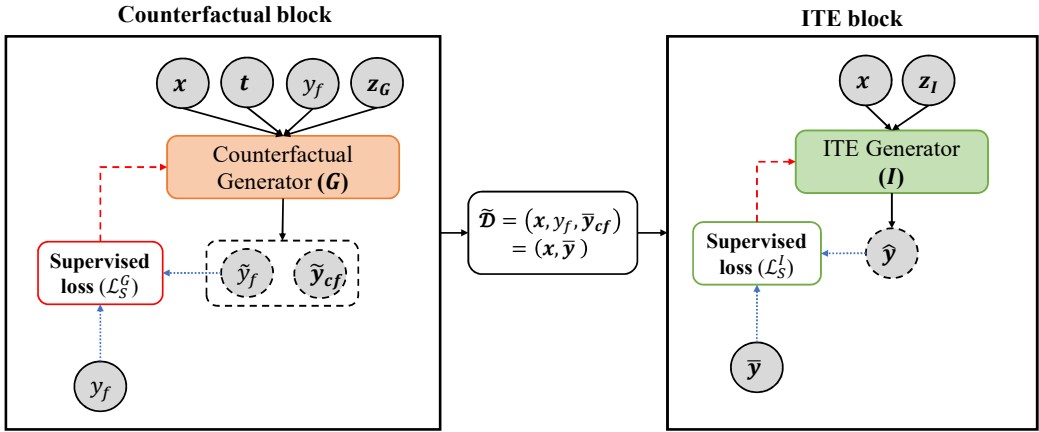

(a) **G:** S loss only, **I:** S loss only

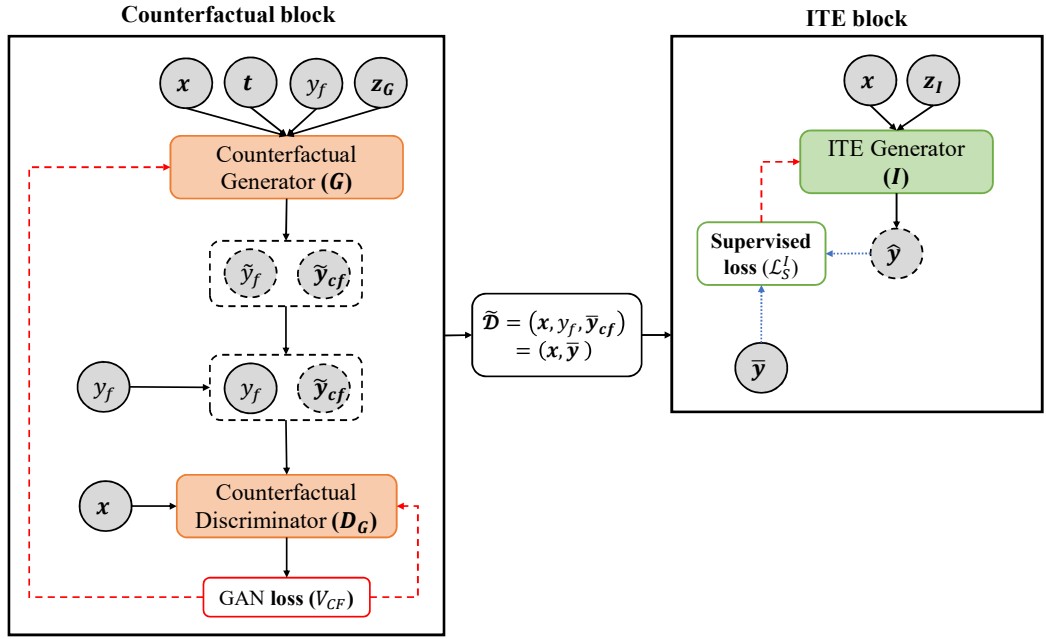

(b) **G:** GAN loss only, **I:** S loss only

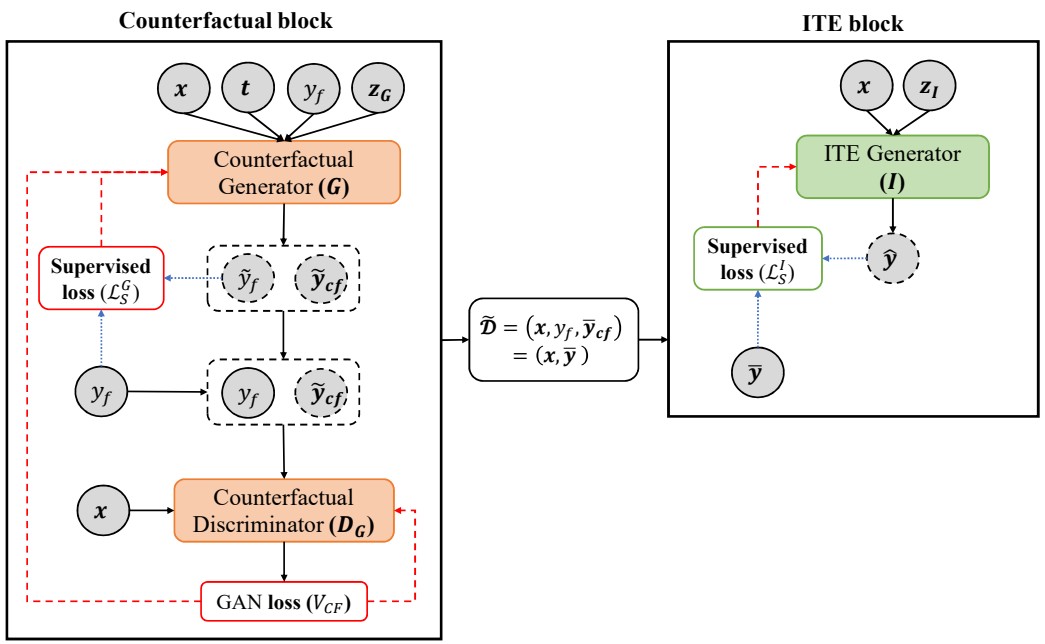

(c) **G:** S and GAN loss, **I:** S loss only

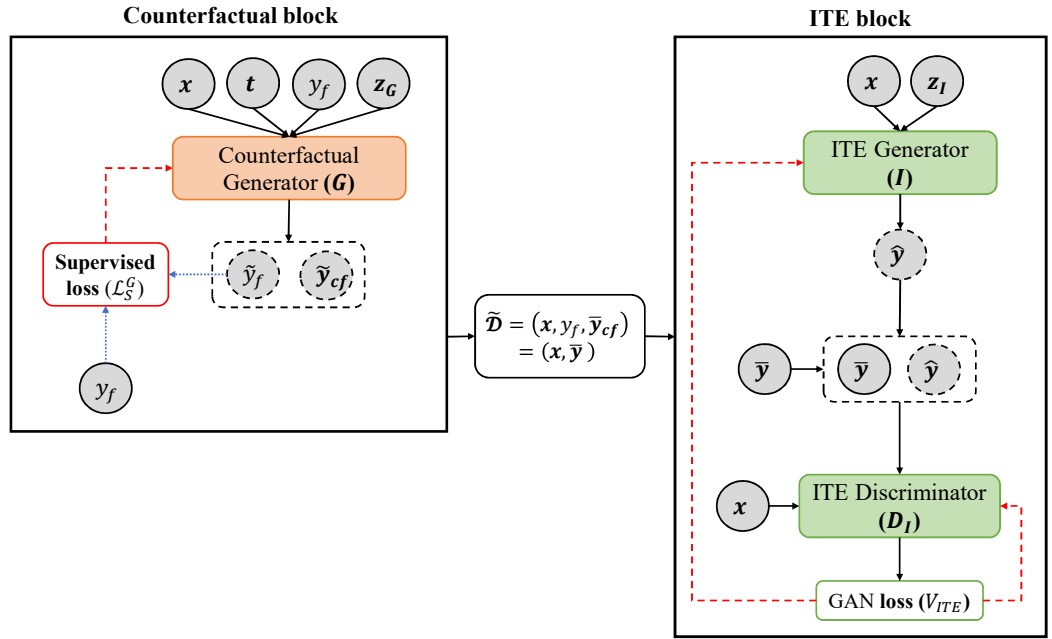

(d) **G:** S loss only, **I:** GAN loss only

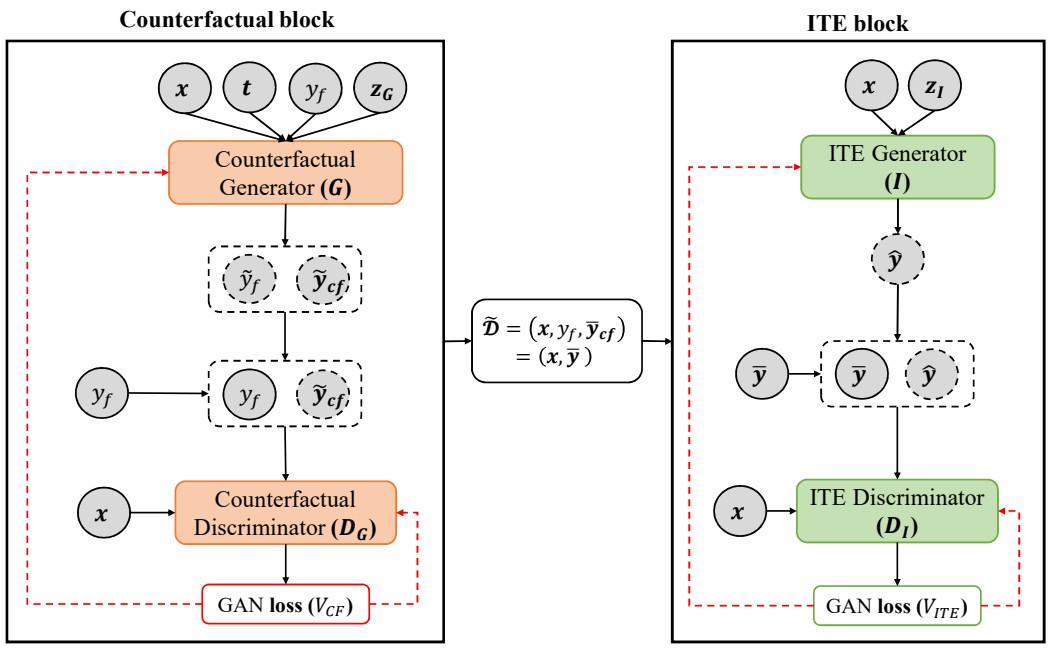

(e) **G:** GAN loss only, **I:** GAN loss only

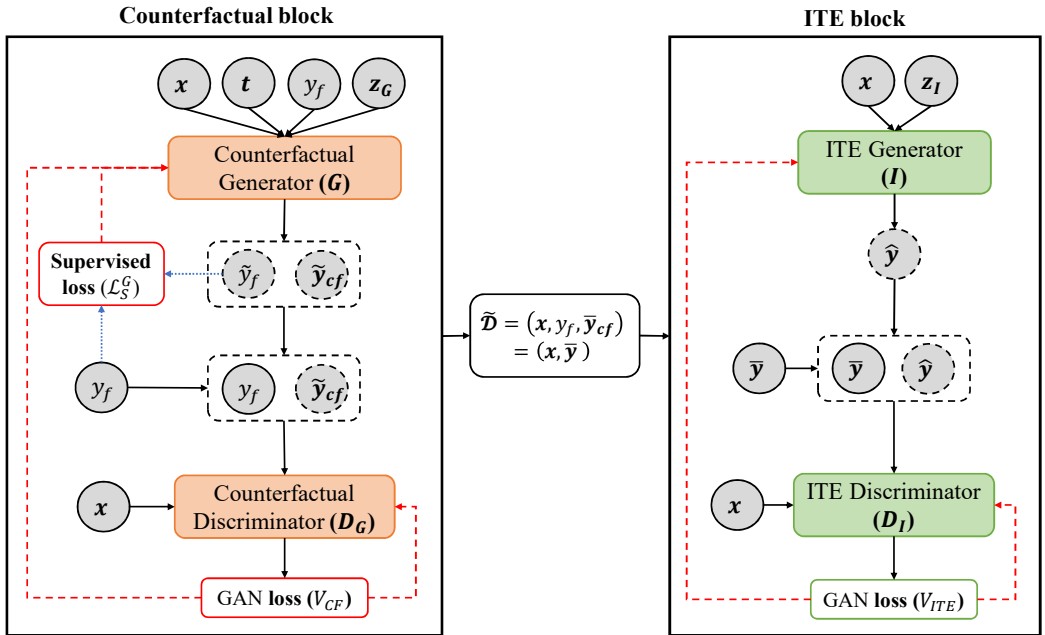

(f) **G:** S and GAN loss, **I:** GAN loss only

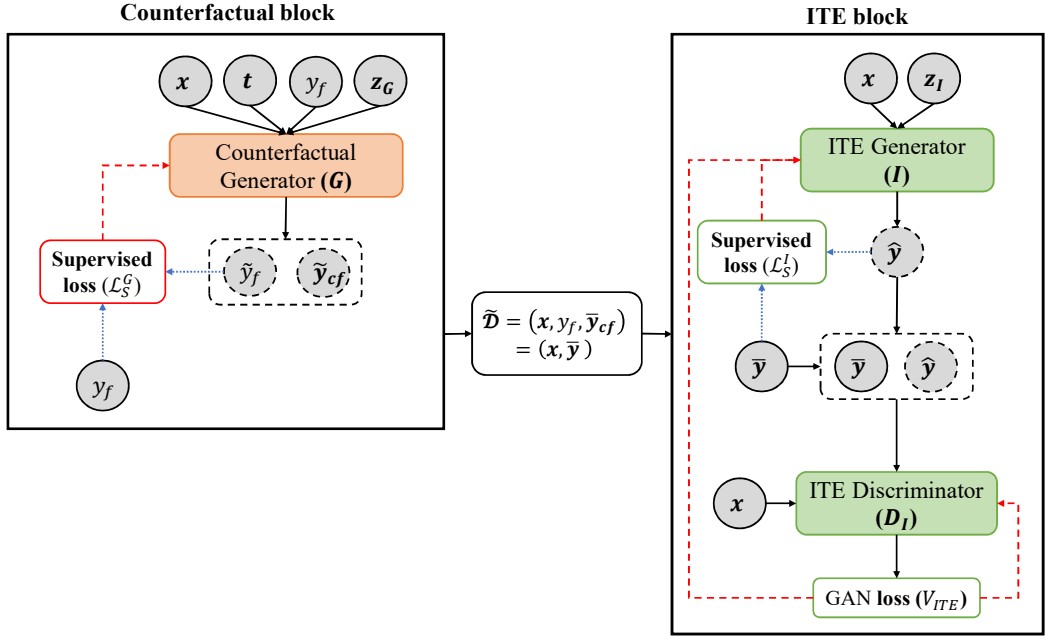

(g) **G:** S loss only, **I:** S and GAN loss

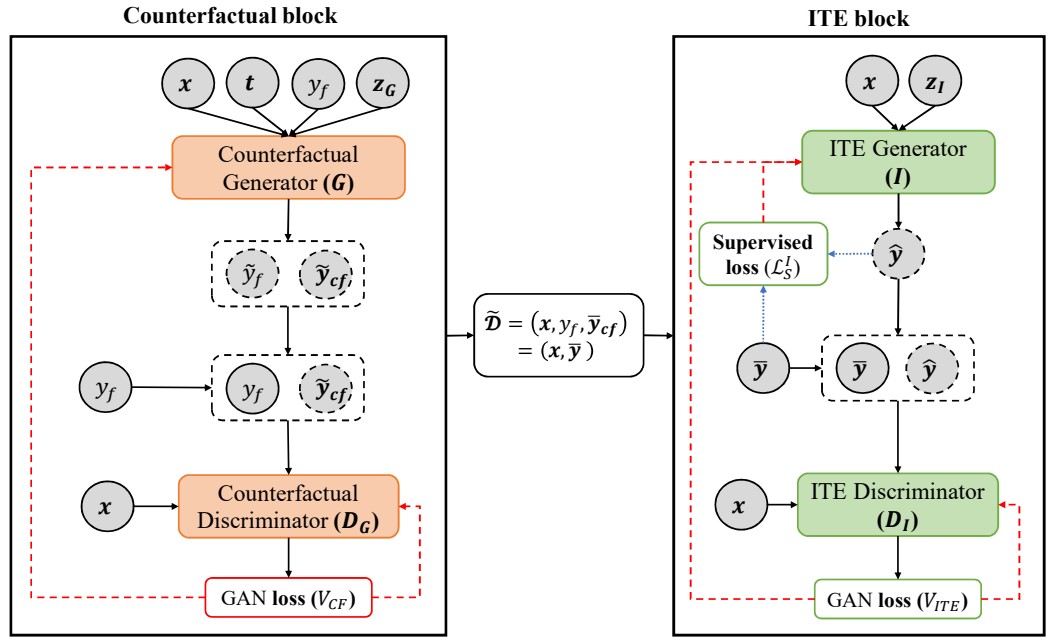

(h) **G:** GAN loss only, **I:** S and GAN loss

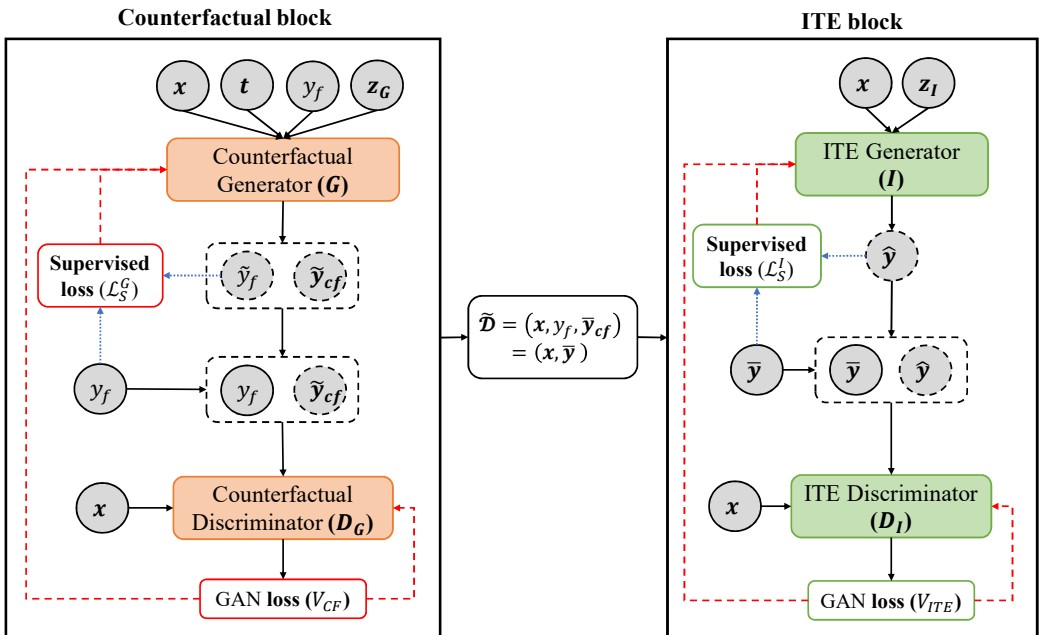

(i) **G:** S and GAN loss**, I:** S and GAN loss

