# OpenReview forum: "GANITE: Estimation of Individualized Treatment Effects using Generative Adversarial Nets"
_ICLR.cc/2018/Conference — Accept (Poster)_

### Official Review · AnonReviewer1 · 2017-11-27
**Good empirical results, the explanation of the methods can be improved**

**Rating:** 6
**Confidence:** 4

**Review:**

Summary:
This paper proposes to estimate the individual treatment effects (ITE) through
training two separate conditional generative adversarial networks (GANs).

First, a counterfactual GAN is trained to estimate the conditional distribution
of the potential outcome vector, which consists of factual outcome and all
other counterfactual outcomes, given 1) the feature vector, 2) the treatment
variable, and 3) the factual outcome. After training the counterfactual GAN,
the complete dataset containing the observed potential outcome vector can be
generated by sampling from its generator.

Second, a ITE GAN is trained to estimate the conditional distribution of the
potential outcome vector given only the feature vector. In this way, for any
test sample, its potential outcomes can be estimated using the generator of the
trained ITE GAN. Given its pontential outcome vector, its ITE can be estimated
as well.

Experimental results on the synthetic data shows the proposed approach, called
GANITE, is more robust to the existence of selection bias, which is defined as
the mismatch between the treated and controlled distributions, compared to
its competing alternatives. Experiments on three real world datasets show
GANITE achieves the best performance on two datasets, including Twins and Jobs.
It does not perform very well on the IHDP dataset. The authors also run
experiments on the Twins dataset to show the proposed approach can estimate the
multiple treatment effects with better performance.

Comments
1) This paper is well written. The background and related works are well
organized.

2) To the best of my knowledge, this is the first work that applies
GAN to ITE estimation.

3) Experiments on the synthetic data and the real-world data demonstrate the
advantage of the proposed approach.

4) The authors directly present the formulation without providing sufficient
motivations. Could the authors provide more details or intuitions on why GAN
would improve the performance of ITE estimation compared to approaches that
learn representations to minimize the distance between the distributions of
different treatment groups, such as CFR_WASS?

5) As is pointed out by the authors, the proposed approach does not perform
well when the dataset is small, such as the IHDP data. However, in practice, a
lot of real-world datasets might have small sample size, such as the LaLonde
dataset. Did the authors plan to extend the model to handle those small-sized
data sets without completely changing the model.

6) When training the ITE GAN, the objective is to learn the conditional
distribution of the potential outcome vector given the feature vector. Did the
authors try the option of replacing ITE GAN with multi-task regression? Will
the performance become worse using multi-task regression?  I think this
comparison would be a sanity check on the utility of using GAN instead of
regression models for ITE estimation.

---

> ### Author Response · Authors · 2017-12-13
> **Re: Good empirical results, the explanation of the methods can be improved**
>
> Answer 1: Inherent to the approach of learning a balanced representation is that the representation must trade off between predictive accuracy and bias. This is because often it will be the case that information that is biased is also highly predictive (in fact in the medical setting this is precisely why it is biased - because the doctors will assign treatments based on predictive features). GANITE on the other hand is not forced to make this bias trade-off and so, as shown in Figure 2, is able to outperform methods such as CFR_WASS, particularly when the bias is high.
>
> A further advantage of GANITE is that PEHE can be estimated from the generated counterfactuals of the “G” network.Therefore, we can directly optimize the hyperparameters that minimize the estimated PEHE which is the performance metric for ITE estimation. On the other hand, existing work such as CFR_WASS [1] cannot directly optimize PEHE.
>
> [1] Shalit, Uri, Fredrik Johansson, and David Sontag. "Estimating individual treatment effect: generalization bounds and algorithms." ICML, 2016.
>
> Answer 2: In the revised manuscript we will show improved results for GANITE on small datasets by searching for hyper-parameters in a larger space.
>
> Answer 3: The ITE GAN can be replaced by any regression method. We show in Table 1 that the ITE GAN outperforms the alternative of replacing it with a multi-layer perceptron (MLP) - this corresponds to Row 1 and Column 3 of Table 1 (S loss only). However, this is not the only reason that we use ITE GAN instead of other regression models. The ITE GAN allows us to estimate the distribution of the potential outcomes, rather than just the expectation, which gives us access to, for example, the variance, capturing the underlying variability of the potential outcomes. We believe this is very important information when a decision about treatments assignments needs to be made [1]. We will try to highlight this more in the revised manuscript.

---

### Official Review · AnonReviewer2 · 2017-11-27
**Novel idea with seemingly a large amount of moving parts and a considerable amount of hyper parameters.**

**Rating:** 6
**Confidence:** 3

**Review:**

This paper presents GANITE, a Generative Adversarial Network (GAN) approach for estimating Individualized Treatment Effects (ITE). This is achieved by utilising a GAN to impute the `missing` counterfactuals, i.e. the  outcomes of the treatments that were not observed in the training (i.e. factual) sample, and then using another GAN to estimate the ITE based on this `complete` dataset. The authors then proceed in combining the two GAN objectives with extra supervised losses to better account for the observed data; the GAN loss for the `G` network has an extra term for the `G` network to better predict the factual outcome `y_f` (which should be easy to do given the fact that y_f is an input to the network) and the GAN loss for the `I` network has an extra term w.r.t. the corresponding performance metric used for evaluation, i.e. PEHE for binary treatment and MSE for multiple treatments. This model is then evaluated on extensive experiments.

The paper is reasonably well-written with clear background and diagrams for the overall architecture. The idea is novel and seems to be relatively effective in practice although I do believe that it has a lot of moving parts and introduces a considerable amount of hyperameters (which generally are problematic to tune in causal inference tasks). Other than that, I have the following questions and remarks:
- I might have misunderstood the motivation but the GAN objective for the `G` network is a bit weird; why is it a good idea to push the counterfactual outcomes close to the factual outcomes (which is what the GAN objective is aiming for)? Intuitively, I would expect that different treatments should have different outcomes and the distribution of the factual and counterfactual `y` should differ.
- According to which metric did you perform hyper-parameter optimization on all of the experiments?
- From the first toy experiment that highlights the importance of each of the losses it seems that the addition of the supervised loss greatly boosts the performance, compared to just using the GAN objectives. What was the relative weighting on those losses in general?
- From what I understand the `I` network is necessary for out-of-sample predictions where you don’t have the treatment assignment, but for within sample prediction you can also use the `G` network. What is the performance gap between the `I` and `G` networks on the within-sample set? Furthermore, have you experimented with constructing `G` in a way that can represent `I` by just zeroing the contribution of `y_f` and `t`? In this way you can tie the parameters and avoid the two-step process (since `G` and `I` represent similar things).
- For figure 2 what was the hyper parameters for CFR? CFR includes a specific knob to account for the larger mismatches between treated and control distributions. Did you do hyper-parameter tuning for all of the methods in this task?
- I would also suggest to not use “between” when referring to the KL-divergence as it is not a symmetric quantity.

Also it should be pointed out that for IHDP the standard evaluation protocol is 1000 replications (rather than 100) so there might be some discrepancy on the scores due to that.

---

> ### Author Response · Authors · 2017-12-13
> **Re: Novel idea with seemingly a large amount of moving parts and a considerable amount of hyper parameters.**
>
> Answer 1: We acknowledge that, due to the lack of ground truth, it is often difficult in causal inference tasks to optimize the hyper-parameters. This is because we never have access to the true loss function (in our case PEHE or MSE) that we are trying to minimize, and so the difficulty arises about what metric to use to tune hyperparameters with respect to, in the absence of the target loss. One of the advantages of GANITE is that our target loss (PEHE or MSE) can be estimated from the generated counterfactuals, unlike other methods such as in [1]. Therefore, we can directly optimize the hyperparameters that minimize this estimated PEHE/MSE - exact details of our hyper-parameter optimization are given in the Appendix. In the revised manuscript, we will state the optimal values for the hyper-parameters that we found for each dataset using greedy search.
>
> [1] Shalit, Uri, Fredrik Johansson, and David Sontag. "Estimating individual treatment effect: generalization bounds and algorithms." ICML, 2016.
>
> Answer 2: We think that our explanation in the manuscript may not have been clear to understand. We will break section 4 into two subsections that explain “G” and “I” networks separately. We agree that different treatments will have different outcomes, however, we think the misunderstanding has come from confusing “factual/counterfactual” with different treatment assignments. It should be noted that “factual” and “counterfactual” do not correspond to specific treatments – for any given sample, it is possible that any treatment is the factual one. It therefore makes sense to try and push counterfactual outcomes from one sample toward the factual outcomes from other (similar) samples for the same treatments. We achieve this by making the objective of “G” to generate counterfactuals in a way that, given the whole vector (factuals and counterfactuals), the discriminator cannot distinguish which element is factual.
>
> We think there may also be some confusion around the “S loss” used for “G”. Due to the structure of “G”, it outputs a full vector of potential outcomes, and so it not only outputs counterfactuals, but also gives a value for the one factual that was used as input. We account for this by using the “S loss” to force the generated outcome for the factual treatment assignment to be close to the factual outcome actual observed. This is because, conditional on observing y_f, the component of y corresponding to y_f should clearly be equal to y_f.
>
> Answer 3: As can be seen in Answer 1, GANITE generates proxies for the counterfactuals and this allows us to estimate the PEHE directly. We minimized the estimated PEHE over the hyper-parameter space. We will clarify this in the revised paper.
>
> Answer 4: In the revised manuscript, we will include the optimal hyper-parameters (alpha and beta) that we found (using greedy search) for each dataset.
>
> Answer 5: The tasks that “G” and “I” perform are fundamentally different. “G” predicts outcomes conditional on features, treatment assignment and outcome of the chosen treatment, (x, t, y_f). “I” predicts outcomes conditional only on the features, x, and so it would be expected for “G” to perform better than “I” when the task is predicting based on (x, t, y_f). This is the only comparison we can make, since “G” is not capable of predicting with only x. Furthermore, “G” and “I” are not at all independent, “I” is trained on a dataset generated by “G” and so “I” is being forced to “fit” to the outcomes “G” has generated. This means that any errors “G” has made in-sample will be pushed forward to “I”.
>
> However, the performance gap between the two will indicate how well “I” is able to learn from “G”, which we believe is an interesting question and so we will add results for this comparison in the Appendix of the revised manuscript.
>
> Answer 6: We did think about this idea. In order to zero out the contribution of “y_f” and “t”, we need to marginalise out the distributions of “y_f” and “t” conditional on “x”, i.e. P(y_f, t | x) because P(y | x) = int P(y | x, y_f, t) P(y_f, t | x). Therefore, in order to zero out “y_f” and “t” we would need to learn P(y_f, t | x). This requires a model to learn and we do not believe this would be any simpler than our proposed structure - in both cases we would still have 2 learning stages.
>
> We will try this idea (zero out the contribution of “y_f” and “t”) and report the results in the Appendix of the revised manuscript.
>
> Answer 7: We follow the code published in https://github.com/clinicalml/cfrnet. We followed the parameter search process suggested in the github using cfr_param_search.py. We will clarify this in the revised manuscript.
>
> Answer 8: We will revise it. Thanks.
>
> Answer 9: For IHDP, we indeed do 1000 replications and report the results. For other experiments (twins and Jobs), we do 100 replications and report the results. We will clarify this in the revised manuscript.

---

### Official Review · AnonReviewer3 · 2017-11-27
**Re: GANITE: Estimation of Individualized Treatment Effects using Generative Adversarial Nets**

**Rating:** 6
**Confidence:** 3

**Review:**

This paper introduces a generative adversarial network (GAN) for estimating individualized treatment effects (ITEs) by (1) learning a generator that tries to fool a discriminator with feature, treatment, potential outcome- vectors, and (2) by learning a GAN for the treatment effect. In my view, the counterfactual component is the interesting and original component, and the results show that the ITE GAN component further improves performance (marginally but not significant). The analysis is conducted on semi-synthetic data sets created to match real data distributions with synthetically introduced selection bias and conducts extensive experimentation. While the results show worse performance compared to existing literature in the experiment with small data sizes, the work does show improvements in larger data sets. However, Table 5 in the appendix suggests these results are not significant when considering average treatment effect estimation (eATE and eATE).

Quality: good. Clarity: acceptable. Originality: original. Significance: marginal.

The ITE GAN does not significantly outperform the counterfactual GAN alone (in the S and GAN loss regime), and in my understanding the counterfactual GAN is the particularly innovative component here, i.e., can the algorithm effectively enough generate indistinguishable counterfactual outcomes from x and noise. I wonder if the paper should focus on this in isolation to better understand and characterize this contribution.

What is the significance of bold in the tables? I'd remove it if it's just to highlight which method is yours.

Discussion section should be called "Conclusion" and a space permitting a Discussion section should be written.
E.g. exploration of the form of the loss when k>2, or when k is exponential e.g. a {0,1}^c hypercube for c potentially related treatment options in an order set.
E.g. implications of underperformance in settings with small data sets. We have lots of large data sets where ground truth is unknown, and relatively more small data sets where we can identify ground truth at some cost.
E.g. discussion of Table 2 (ITEs) where GANITE is outperforming the methods (at least on large data sets) and Table 5 (ATEs) which does not show the same result is warranted. Why might we expect this to the case?

---

> ### Author Response · Authors · 2017-12-13
> **Re: Re: GANITE: Estimation of Individualized Treatment Effects using Generative Adversarial Nets**
>
> Answer 1: We agree with the reviewer’s comment that the most innovative part of our GANITE structure is the counterfactual GAN component. However, as shown in Table 1, using the ITE GAN does improve the PEHE over using just the Counterfactual GAN (See Row 1 Column 3 of Table 1). On top of this, we believe that there is a further novelty in the ITE GAN - using it allows us to estimate the conditional distribution of the potential outcomes, rather than just the expectation. This allows us to estimate the uncertainty in the true distribution of the outcomes which is important to know when deciding which treatments to assign. We will highlight this point in the revised manuscript.
>
> Answer 2: Bold is just to highlight the results of our model. In the revised manuscript, we will use * to highlight statistically significant improvement(s) and remove the bold.
>
> Answer 3: We will do this in the revised manuscript.
>
> Answer 3-1: We agree with the reviewer that this is an interesting discussion point and will add it to the discussion at the end of the revised manuscript.
>
> Answer 3-2: First, we would like to highlight that inherent to this problem is the fact that ground truth is not available, and so in the small datasets in which we can identify the ground truth GANITE is not needed. We therefore believe that what is important is its performance in the large datasets where ground truth is often impossible to identify. We will, however, show improvements for GANITE on smaller datasets by searching for hyper-parameters in a larger space.
>
> Answer 3-3: The problem we address with GANITE is to estimate the ITE. We used the ATE performance as a sanity check for our method - and believe it passes the sanity check, being competitive with most other methods - but do not believe that it is an important metric for distinguishing models where the task is predicting treatment effects on an individual level (and so we only included these results in the Appendix). To highlight why ATE is not a good metric for comparison of ITE methods, we give a simple example. Consider the model that, for each treatment, simply predicts the population mean of the observed outcomes. Then this will often be a highly underfitted model for the task of ITE prediction since there will be many samples that deviate significantly from this mean, however, the ATE will be close to optimal (in this case bias hasn’t actually been accounted for).

---

### Decision · Program_Chairs · 2018-01-29
**ICLR 2018 Conference Acceptance Decision**

**Decision:**

Accept (Poster)

**Comment:**

The reviewers agree that the method is original and mostly well communicated, but have some doubts about the significance of the work.